# Prevalence of Lactose Intolerance in Patients with Hashimoto Thyroiditis and Impact on LT4 Replacement Dose

**DOI:** 10.3390/nu14153017

**Published:** 2022-07-22

**Authors:** Elisa Marabotto, Diego Ferone, Afscin Djahandideh Sheijani, Lara Vera, Sebastiano Ziola, Edoardo Savarino, Giorgia Bodini, Manuele Furnari, Patrizia Zentilin, Vincenzo Savarino, Massimo Giusti, Fabiola Andrea Navarro Rojas, Marcello Bagnasco, Manuela Albertelli, Edoardo G. Giannini

**Affiliations:** 1Division of Gastroenterology, Department of Internal Medicine (DiMI), University of Genoa, IRCCS Ospedale Policlinico San Martino, 16132 Genoa, Italy; elisa.marabotto@unige.it (E.M.); djaf90@hotmail.it (A.D.S.); sebastianoziola@hotmail.it (S.Z.); giorgia.bodini@unige.it (G.B.); manuele.furnari@unige.it (M.F.); pzentilin@unige.it (P.Z.); vsavarin@unige.it (V.S.); egiannini@unige.it (E.G.G.); 2Endocrinology Unit, Department of Internal Medicine and Medical Specialties (DiMI), University of Genoa, IRCCS Ospedale Policlinico San Martino, Viale Benedetto XV, 16132 Genoa, Italy; ferone@unige.it (D.F.); vera.lara79@yahoo.it (L.V.); magius@unige.it (M.G.); fabiola.navarro@gmail.com (F.A.N.R.); bagnasco@unige.it (M.B.); 3Division of Gastroenterology, Department of Surgery, Oncology and Gastroenterology, University of Padua, 35128 Padua, Italy; edoardo.savarino@unipd.it

**Keywords:** lactose intolerance, lactose breath test, hypothyroidism, levothyroxine

## Abstract

Purpose: to determine lactose intolerance (LI) prevalence in women with Hashimoto’s thyroiditis (HT) and assess the impact of LI on LT4 replacement dose. Methods. consecutive patients with HT underwent Lactose Breath Test and clinical/laboratory data collection. Unrelated gastrointestinal disorders were carefully ruled out. Lactose-free diet and shift to lactose-free LT4 were proposed to patients with LI. Results: we enrolled 58 females (age range, 23–72 years) with diagnosis of HT. In total, 15 patients were euthyroid without treatment, and 43 (74%) euthyroid under LT4 (30 of them with a LT4 formulation containing lactose). Gastrointestinal symptoms were present in 84.5% of patients, with a greater prevalence in change in bowel habits in lactose-intolerant patients (*p* < 0.0001). The cumulative LT4 dose required did not differ in patients with or without LI. No significant difference in both TSH values and LT4 dose were observed in patients shifted to lactose-free LT4 and diet at 3 and 6 months compared to baseline. Conclusion: the prevalence of LI in patients with HT was 58.6%, not different from global prevalence of LI. In the absence of other gastrointestinal disorders, LI seems not to be a major cause of LT4 malabsorption and does not affect the LT4 required dose in HT patients.

## 1. Introduction

Autoimmune thyroid diseases (AITD) are the most common autoimmune disorders, have a multifactorial etiology, and develop in the context of a specific genetic background facilitated by exposure to environmental factors [1,2]. Hashimoto’s thyroiditis (HT) is the most common condition, and its diagnosis relies on the demonstration of circulating antibodies to thyroid antigens, anti-thyroperoxidase antibodies [TPOAb], and anti-thyroglobulin antibodies (TgAb), as well as on the reduced echogenicity on thyroid sonogram [3]. In the majority of cases, HT produces hypothyroidism, although at presentation, patients can be euthyroid or even reveal elevated circulating thyroid hormones levels (thyrotoxicosis) [3]. Therapy of hypothyroidism consists in the daily and permanent oral administration of synthetic levothyroxine (LT4) to maintain the thyroid stimulating hormone (TSH) within the normal limits [3]. The absorption of LT4 corresponds to 60–80% of the administered dose and takes place in the intestine, mainly jejunum and ileum [4]. Impaired bioavailability should be suspected in all patients taking more than 2 μg/Kg *per* day, and poor adherence to therapy is a possible cause of treatment failure (pseudo-malabsorption) [5]. However, LT4 absorption can be altered by physiological conditions such as pregnancy, aging, or diet, or by drugs, as well as several pathological conditions affecting the digestive system, such as celiac disease, atrophic gastritis, inflammatory bowel disease, and lactose intolerance [6,7,8,9,10].

Most people are born able to digest lactose, a carbohydrate hydrolyzed into glucose and galactose by the lactase enzyme that is located in the brush border microvilli of the small intestine enterocytes [11,12]. However, after 2–12 years, there are two distinct groups of people, one called “lactase non-persistent” with low lactase activity (hypolactasia), and one called “lactase-persistent”, that include individuals who maintain lactase activity in adulthood [11]. The decreased activity of lactase causes lactose malabsorption, a condition that when symptomatic is called lactose intolerance. Indeed, undigested lactose is fermented in the intestinal lumen by colonic bacteria, leading to production of short-chain fatty acids, hydrogen, carbon dioxide, and methane that cannot be absorbed, thus increasing the osmotic pressure and drawing more fluid into the bowel [13]. These products may lead to gastrointestinal symptoms such as abdominal pain, bloating, flatus, borborygmi, and diarrhea [14,15,16]. Further, the malabsorption that can be a consequence of lactose intolerance might influence the bioavailability of administered drugs, including LT4, due to the presence of lactose in some LT4 formulations [17,18]. As a fact, it was recently demonstrated that lactose intolerance significantly increased the need for oral thyroxine in 34 hypothyroid patients affected by HT and lactose intolerance, noncompliant with a lactose-free diet [18]. A further study by Asik et al. confirmed these findings. However, these latter authors based the diagnosis of lactose intolerance only on the dosage of glucose in the blood after lactose intake [17].

In this study, we prospectively assessed the prevalence of lactose intolerance in a group of consecutive patients with HT, the presence of symptoms associated with intolerance and the benefit of a lactose-free diet and new lactose-free LT4 formulations.

## 2. Patients and Methods

### 2.1. Subjects

This prospective study was conducted in the transition period between the generalized availability of lactose-free LT4 tablet formulations and the persistent use of LT4 formulation containing lactose (available up to autumn 2021), on outpatient basis, and included patients consecutively referred for thyroid diseases to the Endocrinology Unit, University of Genoa. A major inclusion criterion was female patients with diagnosis of HT and aged 18–75 years. We enrolled only female patients due to the very low prevalence of HT in males, who represent no more than 10–25% of HT patients [19]. Exclusion criteria included pregnancy or breast-feeding, use of iodine-containing substances and/or diet creams or pills, use of drugs interfering with levothyroxine absorption and action (in particular, synthetic estrogens), use of drugs interfering with TSH assay, and history of cancer in chemotherapy, gastrointestinal surgery, respiratory disease, autoimmune atrophic gastritis, Crohn’s disease, and celiac disease.

Gastroenterological assessment was carried out on outpatient basis at the Gastroenterology Unit, University of Genoa. All patients signed an informed consent. The study was designed and carried out in accordance with the Helsinki Declaration (Sixth revision, Seoul 2008) and was approved by the local Institutional Review Board.

### 2.2. Protocol

The diagnosis of HT was confirmed by the positivity of thyroid autoantibodies (at least TPOAb > 100 mU/L) and the sonographic thyroid pattern. We recorded patients’ body mass index (BMI kg/m^2^), lifestyle habits (coffee, smoking, alcohol), medications, modality of LT4 assumption, the presence of gastrointestinal symptoms (abdominal pain, change in bowel habits, and bloating), and headache. Morphology and volume of the thyroid were assessed by ultrasound. We tested thyroid function (FT4, TSH) and autoimmunity (TPOAb and TgAb). All patients underwent lactose breath test (LBT) for lactose intolerance and, when indicated, glucose breath test (GBT) was carried out in order to rule out small intestine bacterial overgrowth (SIBO). It is recognized that the gold standard for diagnosing lactose malabsorption is the H2 hydrogen breath test: a diagnostic cut-off of 20 ppm has a reported specificity of 100% at a sensitivity of 60% to identify lactose malabsorption.

Other diagnostic procedures (specific autoantibody testing and endoscopy—see below), in addition to clinical gastroenterological evaluation and basic clinical biochemistry, were carried out to exclude other gastroenterological disorders, such as atrophic gastritis and celiac disease.

The diagnosis of lactose intolerance was based on the presence of gastrointestinal symptoms after milk ingestion and a positive LBT after an accurate evaluation of possible false positive and negative tests. Lactose-free LT4 and a lactose-free diet were proposed to patients with a new diagnosis of lactose intolerance. Patients were then re-evaluated at 3 and 6 months following diet and therapeutic modification.

### 2.3. Endocrinological Assessment

The serum FT4 and TSH were measured by chemiluminescence (Roche Diagnostics, Mannheim, Germany). According to the current consensus recommendations a target TSH level between 0.3–3.5 mIU/L was considered adequate [20,21]. The normality range of FT4 in our laboratory is 7.2–13.2 pmol/L. TPOAb and TgAb were determined by enzyme immunoassay (DRG Instruments GmbH, Marburg, Germany) we considered positive patients those with Ab level > 100 mU/L. Patients were evaluated with ultrasound device MyLab40 with 7.5 MHz linear probe (ESAOTE, Genoa, Italy). Inhomogeneity and hypoechogenecity of the gland, as well as the diffuse pseudonodular appearance and lumps contours were the main sonographic features indicative of chronic thyroiditis [3,22]. Thyroid volume was calculated using the ellipsoid formula determined with the combined volume of the lobes ((D1 × D2 × D3) × 0.52) and expressed in mL [23]. We considered as indicative of thyroid goiter a volume >16 mL.

### 2.4. Gastroenterological Assessment

Gastroenterological evaluation included determination in all patients of anti-transglutaminase antibodies (Anti-tTG, normal values <20 U/mL) (Euroimmun, Luebeck, Germany), total IgA (normal range: 0.7–4.0 g/L), presence of *H. pylori* infection by means of ^13^C-urea breath test (AB Analitica, Padova, Italy), and anti-gastric parietal cell antibodies (APCA, normal titer < 1:80) indirect immunofluorescence assay (Euroimmun, Luebeck, Germany).

The LBT was carried out in the morning according to our previously described protocol [24,25,26]. Anti-secretory and antibiotics were withdrawn 15 days prior to testing. In summary, patients were asked to have a carbohydrate-restricted dinner on the day before the test and to be fasted for at least 12 h on the testing day. Before the test, patients were asked to mouthwash with chlorhexidine. Patients did not smoke and perform physical exercise for 30 min before and during the breath test. A basal sample was collected and a baseline H_2_ value >10 ppm was defined as an exclusion criterion. Any CH_4_ value >5 ppm was used as a cut-off to define patients as “methane producers” [24]. The test consisted in the ingestion of 25 g of lactose in 250 mL of water and in the subsequent measurements of H_2_ and CH_4_ in the expired. A gas chromatograph (Breath Tracker SC, Quintron Instrument co. inc. Milwaukee Milwaukee, USA) every 30 min for 240 min measured H_2_ e CH_4_. LBT was considered diagnostic for lactose intolerance in case of a ΔH_2_ excretion >20 ppm at least 30 min after the intake of lactose [24]. GBT was carried out in order to rule out SIBO [24,26,27]. In case of positive APCA or positive anti-tTG an esophagogastroduodenoscopy was performed to verify the presence of an autoimmune atrophic gastritis or a celiac disease.

### 2.5. Statistical Analysis

Demographic and clinical characteristics of the study population are shown as median and range or as absolute value and percentage, when appropriate. Differences between groups were assessed using Wilcoxon test, when indicated, while the Analysis of Variance (ANOVA) was used to assess TSH at the various study time-points. Percentages were compared by means of χ^2^-test or Fisher’s exact test, when appropriate. For statistical significance, *p* was set at 0.05.

## 3. Results

### 3.1. Patients

The flow of patients in the study is shown in Figure 1. Briefly, we enrolled 58 female Caucasian patients with HT (median age, 45 years; range, 23–72 years). In total, 43 of them had hypothyroidism and were under stable L-thyroxine treatment, and 15 were euthyroid and did not need L-thyroxine. Their demographic, clinical and laboratory data are shown in Table 1. In Table 2 are reported TSH and fT4 levels for the two subgroups: as apparent, in hypothyroid under L-thyroxine and euthyroid patients thyroid function data were superimposable. The diagnosis of HT was present from a median of 5 years (range, 0.5–23 years) before study enrollment, while goiter (thyroid volume > 16 mL) was observed in two patients only (3.4%).

Eight patients (13.8%) showed previously unrecognized positivity for APCA, but only one of them was diagnosed as having atrophic autoimmune gastritis by histology. Only one patient showed positivity for Anti-tTG antibodies, but histologic examination resulted negative for celiac disease.

In total, 49 patients (84.5%) presented gastrointestinal disturbances, and the most frequently reported symptoms were change in bowel habit in 24 (41.4%) and bloating in 29 patients (50.0%), while abdominal pain was present in 15 (25.9%), and headache was reported in 9 (15.5%) patients.

### 3.2. Prevalence of Lactose Intolerance

LBT was positive in 35 patients (60.3%). In one of these patients, the only de novo diagnosed with atrophic gastritis, a secondary cause of lactose malabsorption was observed (GBT positive for SIBO). Therefore, the observed prevalence of lactose intolerance in the study population was 58.6% (n = 34).

Among the gastrointestinal symptoms assessed, we observed a significantly greater prevalence in change in bowel habits in lactose-intolerant patients as compared to non-lactose-intolerant group (70.8 vs. 20.6%, *p* < 0.0001), while abdominal pain (29.0 vs. 23.5%), bloating (50.0 vs. 50.0%), and headache (12.5 vs. 17.7%) were evenly distributed in these study sub-groups (Figure 2). Among the 34 lactose-intolerant patients, 9 were euthyroid (26.5%) and 25 hypothyroid under treatment (73.5%), with a distribution not significantly different as compared to non -lactose intolerant patients (25.0 vs. 75.0%, *p* = 1.0) (Figure 1).

### 3.3. Levothyroxine Treatment

As shown in Table 3, Levothyroxine daily replacement dose required in our cohort was closely similar for non-lactose-intolerant and lactose-intolerant patients. In the subgroup of 25 HT lactose-intolerant patients, 8 patients (32.0%) were already on lactose-free diet and on lactose-free LT4 treatment, while 17 patients (68.0%) were on non-lactose-free LT4, and this latter group was shifted to a lactose-free drug formulation (tablets). Following the institution of a lactose-free diet and a shift to a lactose-free drug formulation, median TSH values at 3 and 6 months did not show any statistically significant modification as compared to baseline (1.32 mIU/L (0.06–7.18) vs. 2.05 mIU/L (1.08–12.6) vs. 1.56 mIU/L (0.55–5.19), *p* = 0.107) (Figure 3) and, therefore, the median dose of LT4 did not require any significant modification (1.37 µg/kg/pc (0.35–2.11) at baseline vs. 1.26 µg/kg/pc (0.68–2.11) at 3 months vs. 1.26 µg/kg/pc (0.68–2.11) at 6 months).

## 4. Discussion

Reaching an optimal TSH target in HT patients can be in some instance challenging, and poor patients’ compliance, the resetting of the hypothalamic-pituitary axis, and a reduced absorption of LT4 from the small intestine could explain the difficulties in fine-tuning treatment encountered in clinical practice. While patients counseling may improve adherence to treatment, identifying potential gastrointestinal causes of drug malabsorption should be thoroughly pursued so as to positively act on these, when feasible.

In this regard, a previous study carried out in a cohort of HT patients demonstrated that lactose-intolerant patients needed a significantly higher dosage of replacement therapy, as compared to their lactose-tolerant counterparts [18]. On another note, a study demonstrated that an 8-week lactose-free diet, with no modification in LT4 dosage, led to a significant decrease in TSH values in a group of hypothyroid patients affected by HT with lactose intolerance [17]. Taken together, the results of these preliminary studies suggested that an early screening for lactose intolerance among HT patient could be recommended in order to identify a possible cause of LT4 malabsorption, and, therefore, to optimize treatment. However, in these studies the fact that the diagnosis of lactose intolerance was based only on the presence of malabsorption, without considering its actual etiology or excluding other common causes, represented a relevant limitation.

In the present study, highly stringent criteria were adopted to rule out other potential causes of malabsorption (both autoimmune and non-autoimmune in nature). Specifically, patients with a diagnosis of associated autoimmune gastritis or celiac disease were excluded, and systematic serological screening with APCA and tTg antibodies was performed, with a prevalence of positivity consistent with what expected [28]. Likewise, the diagnosis of SIBO was adequately ruled out. In these conditions, no difference was observed in the LT4 replacement dose required for lactose intolerant and non-lactose-intolerant hypothyroid HT patients. Again, the observed prevalence of lactose intolerance—identified in a cohort of patients affected by HT—was 58.6%, a figure not much different than global prevalence of lactose intolerance (63–86%), and, in particular, in keeping with data from the healthy population of Mediterranean countries where lower prevalence figures are generally recorded [29]. Moreover, our observation is not different from the global prevalence of lactose intolerance identified using hydrogen breath test (i.e., 57%) [30]. In the literature, only one study previously explored the prevalence of lactose intolerance in patients with HT, demonstrating a higher rate of primary lactose intolerance in Turkish patients affected by HT (i.e., 74.7%). However, in this study, only patients with concomitant celiac disease were excluded, while other potential concomitant confounders, such as SIBO, were not ruled out [17].

Overall, our data and those previously published emphasize the finding that lactose intolerance prevalence is not negligible in patients with HT, and its presence possible should be taken into account. In this regard, even though we observed a high prevalence of gastrointestinal symptoms in patients with HT, a finding consistent both with the prevalence of functional bowel disorders in the female general population and with the expected gastrointestinal complaints in patients with thyroid disease, we observed that a more detailed assessment of gastrointestinal symptoms showed that change in bowel habits was significantly more represented among patients with lactose intolerance, and this finding may guide selectively test patients who report this symptom, rather than screening for lactose intolerance all HT patients [31,32].

Another finding of our study was that no significant modifications were observed in TSH values at 3 and 6 months following the institution of a lactose-free diet and a shift to a lactose-free LT4 formulation, with no need to modify the LT4 dose. This finding, that has relevant clinical reflexes, tends to surmise the fact that the overall thyroid balance is maintained in these patients following adequate treatment, and lactose withdrawal does improve gastrointestinal symptoms, but has little effect on LT4 absorption. In any case, the small amount of lactose in the lactose-containing LT4 formulation previously administered to the patients, far lower than the “threshold” 12 g daily dose, is not sufficient to produce malabsorption, in accordance with what reported by Montalto et al. [33].

The main limitations of this study are represented by the relatively small sample size, and the duration of follow-up, and by the absence of a dedicated quality of life evaluation before and after therapeutic shift. Moreover, a more precise, though cumbersome, two-step approach would have required the introduction of a lactose-free diet alone as first step, followed by endocrinology assessment, and then the shift to a lactose-free LT4 formulation in cases of persistent alterations. However, the results obtained have shown that no significant variation of thyroid status occurred even in the one-step lactose withdrawal approach. The main strength is the careful ruling out of gastrointestinal disorders other than lactose intolerance

In conclusion, our study showed that the prevalence of LI in HT patients is significant and that its presence should be investigated in HT patients with concomitant gastroenterological symptoms, mainly change in bowel habits, in order to recommend a lactose-free diet improving their symptoms and their quality of life. Further, there is little evidence of problems with LT4 replacement in hypothyroid HT patients with LI. Our data highlighted the point that LI seems not affect the LT4 required dosage from our patients and, indeed, cannot be considered a major cause of LT4 malabsorption.

## Figures and Tables

**Figure 1 nutrients-14-03017-f001:**
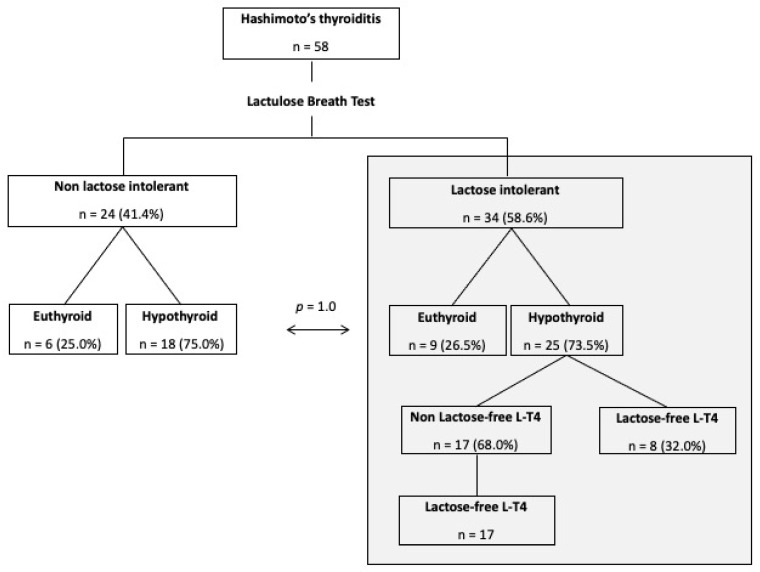
Patients flow within the study.

**Figure 2 nutrients-14-03017-f002:**
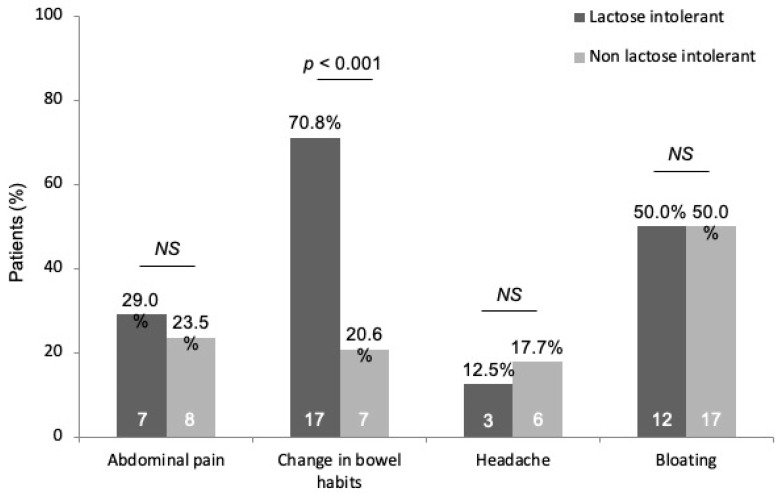
Main symptoms evaluated in the study population, subdivided according to lactose tolerance. Abbreviations: NS, not significant.

**Figure 3 nutrients-14-03017-f003:**
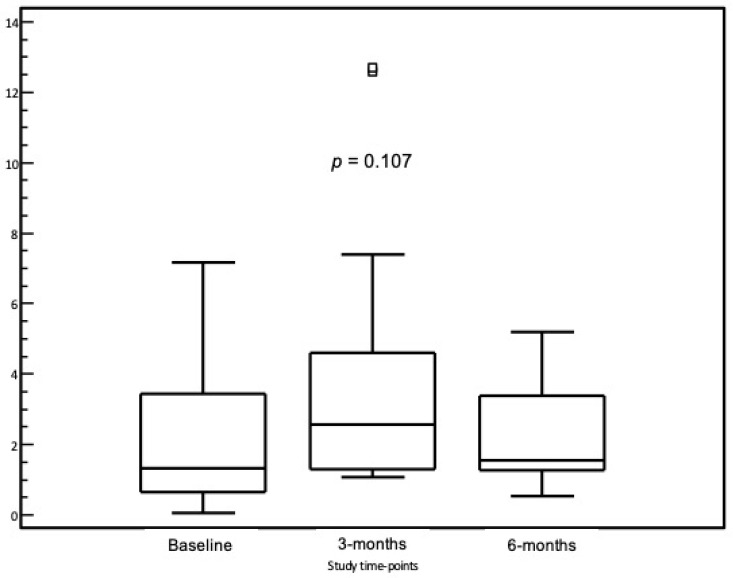
Thyroid-stimulating hormone (TSH) levels in the study cohort following the institution of a lactose-free diet and a shift to a lactose-free drug formulation.

**Table 1 nutrients-14-03017-t001:** Demographic, clinical, and laboratory data of the study population at the baseline.

	Unit	All Patients (n = 58)	Lactose-Intolerant (n = 34)	Non-Lactose-Intolerant (n = 24)
Age	years	45 (23–72)	48 (25–66)	44 (22–72)
Coffee	n (%)	48 (83)	28 (82)	20 (83)
Smokers	n (%)	9 (16)	6 (18)	3 (12.5)
Alcohol	n (%)	26 (45)	17 (50)	9 (37.5)
BMI	kg/m^2^	23.0 (16.9–36.3)	23.0 (16.9–30.4)	22.9 (17.9–36.3)
Thyroid volume >16 mL	n (%)	2 (3.4)	0 (0)	2 (8.3)
TSH level	mIU/L	2.61 (0.06–9.06)	2.61 (0.06–8.82)	2.80 (0.13–9.06)

Data are shown as median and range or as absolute value and percentage. Abbreviations: BMI, Body Mass Index; TSH, Thyroid Stimulating Hormone.

**Table 2 nutrients-14-03017-t002:** Values of fT4, TSH, and LT4 dose of the study population at baseline euthyroid compared to hypothyroid patients under treatment.

	Unit	Euthyroid (n = 15)	Hypothyroid (n = 43)
fT4	pmol/L	10.3 (6.2–14.5)	11.45 (7.8–14.4)
TSH level	mIU/L	2.85 (1.027–6.99)	2.33 (0.06–9.06)
LT4	mcg/kg/die	-	1.13 (0.35–2.46)

Data are shown as median and range. Abbreviations: fT4, free thyroxine; LT4, Levothyroxine; TSH, Thyroid Stimulating Hormone.

**Table 3 nutrients-14-03017-t003:** Values of fT4, TSH, and LT4 dose in the study population at baseline comparing non-lactose-intolerant and lactose-intolerant patients.

	Unit	Hypothyroid non LI (n = 18)	Hypothyroid LI (n = 25)
fT4	pmol/L	11.70 (9.2–14.3)	11.40 (7.8–14.4)
TSH	mIU/L	3.56 (0.06–9.06)	2.16 (0.06–8.82)
L-T4	mcg/kg/die	1.13 (0.54–2.46)	1.05 (0.35–2.11)

Data are shown as median and range. Abbreviations: fT4, free thyroxine; LT4, Levothyroxine; TSH, Thyroid Stimulating Hormone; LI, lactose intolerant.

## Data Availability

We exclude this statement because is not applicable to this study.

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
