# Peer review of "Prevalence of Lactose Intolerance in Patients with Hashimoto Thyroiditis and Impact on LT4 Replacement Dose"

_nutrients, 2022, doi:10.3390/nu14153017_

Round 1

Reviewer 1 Report

The authors present results of their study, where 58 patients with HT (43 of whom were taking thyroxine) were systematically screened for lactose intolerance by lactose breath testing. Importantly, coeliac disease and atrophic gastritis were excluded. In contrast to some previous studies, the authors did not find that a higher thyroxine dose was required in lactose intolerant individuals, nor that changing to a lactose free brand of thyroxine resulted in a change in TSH level. I found the study interesting, well conducted and clinically relevant. 

Major comments

1.     Please make it clear what percentage of patients had SIBO testing. I am under the impression that all 58 patients had anti-tTG and parietal cell antibodies checked- please state this specifically. 

2.     In the title, change “Hashimoto” to “Hasimoto’s”

3. Could the authors comment on tests used to diagnose lactose intolerance and their relative merit? Which test is the "gold standard"?

Minor comments

1.     Suggest change the word “framework” in introduction (“Hashimoto’s thyroiditis is the most common framework”)

2.     Endoscopic evaluation was performed if the autoantibody testing for either coeliac disease or atrophic gastritis was positive. What is the likelihood that a patient may have atrophic gastritis with negative parietal cell antibodies?

3.     Suggest change “non negligible” in final paragraph. 

Reviewer 2 Report

Although relatively small sample size, the study is well constructed and useful to the reader managing patients with Hashimoto thyroiditis.

- the title should reflect the conclusion, which is the absence of impact of lactose intolerance

- the conclusion should include advice for the clinician around the role utility of breath tests and dietary exclusions in these patients

Reviewer 3 Report

The authors have organized this study with the aim of determining the prevalence of lactose intolerance (LI) in 58 patients with Hashimoto’s thyroiditis and assessing the impact of LI on LT4 replacement dose. The cumulative required LT4 dose was not different in patients with or without LI and no significant difference in either TSH values or LT4 dose was observed. The prevalence of LI in patients with HT was 58.6% and did not differ from the global prevalence of LI.

It is a carefully designed and well-presented study. However, I have some concerns:

- Patients with difficulty in controlling their hypothyroidism have more frequent gastrointestinal disorders (GIDs) (McMilan M Drugs RD 2016). The prevalence of LI in a much larger study including 925 patients treated with LT4 was only 7.8%, while gastroesophageal reflux was more frequent, at 33.8%. The study stresses the need for gastrointestinal workup for patients with LT4 malabsorption. Many patients may have other GIDs in addition to LI, such as inflammatory bowel syndrome, which may interfere with LT4 absorption:  this aspect could be more emphatically stated in the study, which shows that nonpersistent LI is not likely to be the cause of LT4 malabsorption, but other underlying GIDs.

- Did the authors detect any difference in LT4 absorption between patients with persistent and nonpersistent lactose intolerance? Munoz-Torres M et al. Thyroid.

- Were all patients treated with LT4 in tablets or else gel formulations?  Some more references can be added: Marek Ruchała et al. Endokrynol Pol 2012; Asik M, Gunes F, et al. Endocrine 2014.

- Of note, intolerance to lactose should be considered in the DD of GIDs that can cause malabsorption of LT (4). The possibility of correcting this disorder with simple dietary measures justifies its consideration.

The results of this study are partially discordant with those of several other studies, as correctly stated by the authors, these discrepancies are due to the different methodologies used and numbers of patients recruited and not to the ascertained diagnosis.  Despite the limited number of study patients, the authors may tentatively suggest guidance on the basis of their results. The latter should be included in the discussion and in the abstract, this highly likely to increase the visibility of the article and of the journal.

The authors have organized this study with the aim of determining the prevalence of lactose intolerance (LI) in 58 patients with Hashimoto’s thyroiditis and assessing the impact of LI on LT4 replacement dose. The cumulative required LT4 dose was not different in patients with or without LI and no significant difference in either TSH values or LT4 dose was observed. The prevalence of LI in patients with HT was 58.6% and did not differ from the global prevalence of LI.

It is a carefully designed and well-presented study. However, I have some concerns:

- Patients with difficulty in controlling their hypothyroidism have more frequent gastrointestinal disorders (GIDs) (McMilan M Drugs RD 2016). The prevalence of LI in a much larger study including 925 patients treated with LT4 was only 7.8%, while gastroesophageal reflux was more frequent, at 33.8%. The study stresses the need for gastrointestinal workup for patients with LT4 malabsorption. Many patients may have other GIDs in addition to LI, such as inflammatory bowel syndrome, which may interfere with LT4 absorption:  this aspect could be more emphatically stated in the study, which shows that nonpersistent LI is not likely to be the cause of LT4 malabsorption, but other underlying GIDs.

- Did the authors detect any difference in LT4 absorption between patients with persistent and nonpersistent lactose intolerance? Munoz-Torres M et al. Thyroid.

- Were all patients treated with LT4 in tablets or else gel formulations?  Some more references can be added: Marek Ruchała et al. Endokrynol Pol 2012; Asik M, Gunes F, et al. Endocrine 2014.

- Of note, intolerance to lactose should be considered in the DD of GIDs that can cause malabsorption of LT (4). The possibility of correcting this disorder with simple dietary measures justifies its consideration.

The results of this study are partially discordant with those of several other studies, as correctly stated by the authors, these discrepancies are due to the different methodologies used and numbers of patients recruited and not to the ascertained diagnosis.  Despite the limited number of study patients, the authors may tentatively suggest guidance on the basis of their results. The latter should be included in the discussion and in the abstract, this highly likely to increase the visibility of the article and of the journal.

The authors have organized this study with the aim of determining the prevalence of lactose intolerance (LI) in 58 patients with Hashimoto’s thyroiditis and assessing the impact of LI on LT4 replacement dose. The cumulative required LT4 dose was not different in patients with or without LI and no significant difference in either TSH values or LT4 dose was observed. The prevalence of LI in patients with HT was 58.6% and did not differ from the global prevalence of LI.

It is a carefully designed and well-presented study. However, I have some concerns:

- Patients with difficulty in controlling their hypothyroidism have more frequent gastrointestinal disorders (GIDs) (McMilan M Drugs RD 2016). The prevalence of LI in a much larger study including 925 patients treated with LT4 was only 7.8%, while gastroesophageal reflux was more frequent, at 33.8%. The study stresses the need for gastrointestinal workup for patients with LT4 malabsorption. Many patients may have other GIDs in addition to LI, such as inflammatory bowel syndrome, which may interfere with LT4 absorption:  this aspect could be more emphatically stated in the study, which shows that nonpersistent LI is not likely to be the cause of LT4 malabsorption, but other underlying GIDs.

- Did the authors detect any difference in LT4 absorption between patients with persistent and nonpersistent lactose intolerance? Munoz-Torres M et al. Thyroid.

- Were all patients treated with LT4 in tablets or else gel formulations?  Some more references can be added: Marek Ruchała et al. Endokrynol Pol 2012; Asik M, Gunes F, et al. Endocrine 2014.

- Of note, intolerance to lactose should be considered in the DD of GIDs that can cause malabsorption of LT (4). The possibility of correcting this disorder with simple dietary measures justifies its consideration.

The results of this study are partially discordant with those of several other studies, as correctly stated by the authors, these discrepancies are due to the different methodologies used and numbers of patients recruited and not to the ascertained diagnosis.  Despite the limited number of study patients, the authors may tentatively suggest guidance on the basis of their results. The latter should be included in the discussion and in the abstract, this highly likely to increase the visibility of the article and of the journal.

The authors have organized this study with the aim of determining the prevalence of lactose intolerance (LI) in 58 patients with Hashimoto’s thyroiditis and assessing the impact of LI on LT4 replacement dose. The cumulative required LT4 dose was not different in patients with or without LI and no significant difference in either TSH values or LT4 dose was observed. The prevalence of LI in patients with HT was 58.6% and did not differ from the global prevalence of LI.

It is a carefully designed and well-presented study. However, I have some concerns:

- Patients with difficulty in controlling their hypothyroidism have more frequent gastrointestinal disorders (GIDs) (McMilan M Drugs RD 2016). The prevalence of LI in a much larger study including 925 patients treated with LT4 was only 7.8%, while gastroesophageal reflux was more frequent, at 33.8%. The study stresses the need for gastrointestinal workup for patients with LT4 malabsorption. Many patients may have other GIDs in addition to LI, such as inflammatory bowel syndrome, which may interfere with LT4 absorption:  this aspect could be more emphatically stated in the study, which shows that nonpersistent LI is not likely to be the cause of LT4 malabsorption, but other underlying GIDs.

- Did the authors detect any difference in LT4 absorption between patients with persistent and nonpersistent lactose intolerance? Munoz-Torres M et al. Thyroid.

- Were all patients treated with LT4 in tablets or else gel formulations?  Some more references can be added: Marek Ruchała et al. Endokrynol Pol 2012; Asik M, Gunes F, et al. Endocrine 2014.

- Of note, intolerance to lactose should be considered in the DD of GIDs that can cause malabsorption of LT (4). The possibility of correcting this disorder with simple dietary measures justifies its consideration.

The results of this study are partially discordant with those of several other studies, as correctly stated by the authors, these discrepancies are due to the different methodologies used and numbers of patients recruited and not to the ascertained diagnosis.  Despite the limited number of study patients, the authors may tentatively suggest guidance on the basis of their results. The latter should be included in the discussion and in the abstract, this highly likely to increase the visibility of the article and of the journal.

The authors have organized this study with the aim of determining the prevalence of lactose intolerance (LI) in 58 patients with Hashimoto’s thyroiditis and assessing the impact of LI on LT4 replacement dose. The cumulative required LT4 dose was not different in patients with or without LI and no significant difference in either TSH values or LT4 dose was observed. The prevalence of LI in patients with HT was 58.6% and did not differ from the global prevalence of LI.

It is a carefully designed and well-presented study. However, I have some concerns:

- Patients with difficulty in controlling their hypothyroidism have more frequent gastrointestinal disorders (GIDs) (McMilan M Drugs RD 2016). The prevalence of LI in a much larger study including 925 patients treated with LT4 was only 7.8%, while gastroesophageal reflux was more frequent, at 33.8%. The study stresses the need for gastrointestinal workup for patients with LT4 malabsorption. Many patients may have other GIDs in addition to LI, such as inflammatory bowel syndrome, which may interfere with LT4 absorption:  this aspect could be more emphatically stated in the study, which shows that nonpersistent LI is not likely to be the cause of LT4 malabsorption, but other underlying GIDs.

- Did the authors detect any difference in LT4 absorption between patients with persistent and nonpersistent lactose intolerance? Munoz-Torres M et al. Thyroid.

- Were all patients treated with LT4 in tablets or else gel formulations?  Some more references can be added: Marek Ruchała et al. Endokrynol Pol 2012; Asik M, Gunes F, et al. Endocrine 2014.

- Of note, intolerance to lactose should be considered in the DD of GIDs that can cause malabsorption of LT (4). The possibility of correcting this disorder with simple dietary measures justifies its consideration.

The results of this study are partially discordant with those of several other studies, as correctly stated by the authors, these discrepancies are due to the different methodologies used and numbers of patients recruited and not to the ascertained diagnosis.  Despite the limited number of study patients, the authors may tentatively suggest guidance on the basis of their results. The latter should be included in the discussion and in the abstract, this highly likely to increase the visibility of the article and of the journal.

The authors have organized this study with the aim of determining the prevalence of lactose intolerance (LI) in 58 patients with Hashimoto’s thyroiditis and assessing the impact of LI on LT4 replacement dose. The cumulative required LT4 dose was not different in patients with or without LI and no significant difference in either TSH values or LT4 dose was observed. The prevalence of LI in patients with HT was 58.6% and did not differ from the global prevalence of LI.

It is a carefully designed and well-presented study. However, I have some concerns:

- Patients with difficulty in controlling their hypothyroidism have more frequent gastrointestinal disorders (GIDs) (McMilan M Drugs RD 2016). The prevalence of LI in a much larger study including 925 patients treated with LT4 was only 7.8%, while gastroesophageal reflux was more frequent, at 33.8%. The study stresses the need for gastrointestinal workup for patients with LT4 malabsorption. Many patients may have other GIDs in addition to LI, such as inflammatory bowel syndrome, which may interfere with LT4 absorption:  this aspect could be more emphatically stated in the study, which shows that nonpersistent LI is not likely to be the cause of LT4 malabsorption, but other underlying GIDs.

- Did the authors detect any difference in LT4 absorption between patients with persistent and nonpersistent lactose intolerance? Munoz-Torres M et al. Thyroid.

- Were all patients treated with LT4 in tablets or else gel formulations?  Some more references can be added: Marek Ruchała et al. Endokrynol Pol 2012; Asik M, Gunes F, et al. Endocrine 2014.

- Of note, intolerance to lactose should be considered in the DD of GIDs that can cause malabsorption of LT (4). The possibility of correcting this disorder with simple dietary measures justifies its consideration.

The results of this study are partially discordant with those of several other studies, as correctly stated by the authors, these discrepancies are due to the different methodologies used and numbers of patients recruited and not to the ascertained diagnosis.  Despite the limited number of study patients, the authors may tentatively suggest guidance on the basis of their results. The latter should be included in the discussion and in the abstract, this highly likely to increase the visibility of the article and of the journal.

The authors have organized this study with the aim of determining the prevalence of lactose intolerance (LI) in 58 patients with Hashimoto’s thyroiditis and assessing the impact of LI on LT4 replacement dose. The cumulative required LT4 dose was not different in patients with or without LI and no significant difference in either TSH values or LT4 dose was observed. The prevalence of LI in patients with HT was 58.6% and did not differ from the global prevalence of LI.

It is a carefully designed and well-presented study. However, I have some concerns:

- Patients with difficulty in controlling their hypothyroidism have more frequent gastrointestinal disorders (GIDs) (McMilan M Drugs RD 2016). The prevalence of LI in a much larger study including 925 patients treated with LT4 was only 7.8%, while gastroesophageal reflux was more frequent, at 33.8%. The study stresses the need for gastrointestinal workup for patients with LT4 malabsorption. Many patients may have other GIDs in addition to LI, such as inflammatory bowel syndrome, which may interfere with LT4 absorption:  this aspect could be more emphatically stated in the study, which shows that nonpersistent LI is not likely to be the cause of LT4 malabsorption, but other underlying GIDs.

- Did the authors detect any difference in LT4 absorption between patients with persistent and nonpersistent lactose intolerance? Munoz-Torres M et al. Thyroid.

- Were all patients treated with LT4 in tablets or else gel formulations?  Some more references can be added: Marek Ruchała et al. Endokrynol Pol 2012; Asik M, Gunes F, et al. Endocrine 2014.

- Of note, intolerance to lactose should be considered in the DD of GIDs that can cause malabsorption of LT (4). The possibility of correcting this disorder with simple dietary measures justifies its consideration.

The results of this study are partially discordant with those of several other studies, as correctly stated by the authors, these discrepancies are due to the different methodologies used and numbers of patients recruited and not to the ascertained diagnosis.  Despite the limited number of study patients, the authors may tentatively suggest guidance on the basis of their results. The latter should be included in the discussion and in the abstract, this highly likely to increase the visibility of the article and of the journal.

The authors have organized this study with the aim of determining the prevalence of lactose intolerance (LI) in 58 patients with Hashimoto’s thyroiditis and assessing the impact of LI on LT4 replacement dose. The cumulative required LT4 dose was not different in patients with or without LI and no significant difference in either TSH values or LT4 dose was observed. The prevalence of LI in patients with HT was 58.6% and did not differ from the global prevalence of LI.

It is a carefully designed and well-presented study. However, I have some concerns:

- Patients with difficulty in controlling their hypothyroidism have more frequent gastrointestinal disorders (GIDs) (McMilan M Drugs RD 2016). The prevalence of LI in a much larger study including 925 patients treated with LT4 was only 7.8%, while gastroesophageal reflux was more frequent, at 33.8%. The study stresses the need for gastrointestinal workup for patients with LT4 malabsorption. Many patients may have other GIDs in addition to LI, such as inflammatory bowel syndrome, which may interfere with LT4 absorption:  this aspect could be more emphatically stated in the study, which shows that nonpersistent LI is not likely to be the cause of LT4 malabsorption, but other underlying GIDs.

- Did the authors detect any difference in LT4 absorption between patients with persistent and nonpersistent lactose intolerance? Munoz-Torres M et al. Thyroid.

- Were all patients treated with LT4 in tablets or else gel formulations?  Some more references can be added: Marek Ruchała et al. Endokrynol Pol 2012; Asik M, Gunes F, et al. Endocrine 2014.

- Of note, intolerance to lactose should be considered in the DD of GIDs that can cause malabsorption of LT (4). The possibility of correcting this disorder with simple dietary measures justifies its consideration.

The results of this study are partially discordant with those of several other studies, as correctly stated by the authors, these discrepancies are due to the different methodologies used and numbers of patients recruited and not to the ascertained diagnosis.  Despite the limited number of study patients, the authors may tentatively suggest guidance on the basis of their results. The latter should be included in the discussion and in the abstract, this highly likely to increase the visibility of the article and of the journal.

The authors have organized this study with the aim of determining the prevalence of lactose intolerance (LI) in 58 patients with Hashimoto’s thyroiditis and assessing the impact of LI on LT4 replacement dose. The cumulative required LT4 dose was not different in patients with or without LI and no significant difference in either TSH values or LT4 dose was observed. The prevalence of LI in patients with HT was 58.6% and did not differ from the global prevalence of LI.

It is a carefully designed and well-presented study. However, I have some concerns:

- Patients with difficulty in controlling their hypothyroidism have more frequent gastrointestinal disorders (GIDs) (McMilan M Drugs RD 2016). The prevalence of LI in a much larger study including 925 patients treated with LT4 was only 7.8%, while gastroesophageal reflux was more frequent, at 33.8%. The study stresses the need for gastrointestinal workup for patients with LT4 malabsorption. Many patients may have other GIDs in addition to LI, such as inflammatory bowel syndrome, which may interfere with LT4 absorption:  this aspect could be more emphatically stated in the study, which shows that nonpersistent LI is not likely to be the cause of LT4 malabsorption, but other underlying GIDs.

- Did the authors detect any difference in LT4 absorption between patients with persistent and nonpersistent lactose intolerance? Munoz-Torres M et al. Thyroid.

- Were all patients treated with LT4 in tablets or else gel formulations?  Some more references can be added: Marek Ruchała et al. Endokrynol Pol 2012; Asik M, Gunes F, et al. Endocrine 2014.

- Of note, intolerance to lactose should be considered in the DD of GIDs that can cause malabsorption of LT (4). The possibility of correcting this disorder with simple dietary measures justifies its consideration.

The results of this study are partially discordant with those of several other studies, as correctly stated by the authors, these discrepancies are due to the different methodologies used and numbers of patients recruited and not to the ascertained diagnosis.  Despite the limited number of study patients, the authors may tentatively suggest guidance on the basis of their results. The latter should be included in the discussion and in the abstract, this highly likely to increase the visibility of the article and of the journal.

The authors have organized this study with the aim of determining the prevalence of lactose intolerance (LI) in 58 patients with Hashimoto’s thyroiditis and assessing the impact of LI on LT4 replacement dose. The cumulative required LT4 dose was not different in patients with or without LI and no significant difference in either TSH values or LT4 dose was observed. The prevalence of LI in patients with HT was 58.6% and did not differ from the global prevalence of LI.

It is a carefully designed and well-presented study. However, I have some concerns:

- Patients with difficulty in controlling their hypothyroidism have more frequent gastrointestinal disorders (GIDs) (McMilan M Drugs RD 2016). The prevalence of LI in a much larger study including 925 patients treated with LT4 was only 7.8%, while gastroesophageal reflux was more frequent, at 33.8%. The study stresses the need for gastrointestinal workup for patients with LT4 malabsorption. Many patients may have other GIDs in addition to LI, such as inflammatory bowel syndrome, which may interfere with LT4 absorption:  this aspect could be more emphatically stated in the study, which shows that nonpersistent LI is not likely to be the cause of LT4 malabsorption, but other underlying GIDs.

- Did the authors detect any difference in LT4 absorption between patients with persistent and nonpersistent lactose intolerance? Munoz-Torres M et al. Thyroid.

- Were all patients treated with LT4 in tablets or else gel formulations?  Some more references can be added: Marek Ruchała et al. Endokrynol Pol 2012; Asik M, Gunes F, et al. Endocrine 2014.

- Of note, intolerance to lactose should be considered in the DD of GIDs that can cause malabsorption of LT (4). The possibility of correcting this disorder with simple dietary measures justifies its consideration.

The results of this study are partially discordant with those of several other studies, as correctly stated by the authors, these discrepancies are due to the different methodologies used and numbers of patients recruited and not to the ascertained diagnosis.  Despite the limited number of study patients, the authors may tentatively suggest guidance on the basis of their results. The latter should be included in the discussion and in the abstract, this highly likely to increase the visibility of the article and of the journal.

The authors have organized this study with the aim of determining the prevalence of lactose intolerance (LI) in 58 patients with Hashimoto’s thyroiditis and assessing the impact of LI on LT4 replacement dose. The cumulative required LT4 dose was not different in patients with or without LI and no significant difference in either TSH values or LT4 dose was observed. The prevalence of LI in patients with HT was 58.6% and did not differ from the global prevalence of LI.

It is a carefully designed and well-presented study. However, I have some concerns:

- Patients with difficulty in controlling their hypothyroidism have more frequent gastrointestinal disorders (GIDs) (McMilan M Drugs RD 2016). The prevalence of LI in a much larger study including 925 patients treated with LT4 was only 7.8%, while gastroesophageal reflux was more frequent, at 33.8%. The study stresses the need for gastrointestinal workup for patients with LT4 malabsorption. Many patients may have other GIDs in addition to LI, such as inflammatory bowel syndrome, which may interfere with LT4 absorption:  this aspect could be more emphatically stated in the study, which shows that nonpersistent LI is not likely to be the cause of LT4 malabsorption, but other underlying GIDs.

- Did the authors detect any difference in LT4 absorption between patients with persistent and nonpersistent lactose intolerance? Munoz-Torres M et al. Thyroid.

- Were all patients treated with LT4 in tablets or else gel formulations?  Some more references can be added: Marek Ruchała et al. Endokrynol Pol 2012; Asik M, Gunes F, et al. Endocrine 2014.

- Of note, intolerance to lactose should be considered in the DD of GIDs that can cause malabsorption of LT (4). The possibility of correcting this disorder with simple dietary measures justifies its consideration.

The results of this study are partially discordant with those of several other studies, as correctly stated by the authors, these discrepancies are due to the different methodologies used and numbers of patients recruited and not to the ascertained diagnosis.  Despite the limited number of study patients, the authors may tentatively suggest guidance on the basis of their results. The latter should be included in the discussion and in the abstract, this highly likely to increase the visibility of the article and of the journal.

The authors have organized this study with the aim of determining the prevalence of lactose intolerance (LI) in 58 patients with Hashimoto’s thyroiditis and assessing the impact of LI on LT4 replacement dose. The cumulative required LT4 dose was not different in patients with or without LI and no significant difference in either TSH values or LT4 dose was observed. The prevalence of LI in patients with HT was 58.6% and did not differ from the global prevalence of LI.

It is a carefully designed and well-presented study. However, I have some concerns:

- Patients with difficulty in controlling their hypothyroidism have more frequent gastrointestinal disorders (GIDs) (McMilan M Drugs RD 2016). The prevalence of LI in a much larger study including 925 patients treated with LT4 was only 7.8%, while gastroesophageal reflux was more frequent, at 33.8%. The study stresses the need for gastrointestinal workup for patients with LT4 malabsorption. Many patients may have other GIDs in addition to LI, such as inflammatory bowel syndrome, which may interfere with LT4 absorption:  this aspect could be more emphatically stated in the study, which shows that nonpersistent LI is not likely to be the cause of LT4 malabsorption, but other underlying GIDs.

- Did the authors detect any difference in LT4 absorption between patients with persistent and nonpersistent lactose intolerance? Munoz-Torres M et al. Thyroid.

- Were all patients treated with LT4 in tablets or else gel formulations?  Some more references can be added: Marek Ruchała et al. Endokrynol Pol 2012; Asik M, Gunes F, et al. Endocrine 2014.

- Of note, intolerance to lactose should be considered in the DD of GIDs that can cause malabsorption of LT (4). The possibility of correcting this disorder with simple dietary measures justifies its consideration.

The results of this study are partially discordant with those of several other studies, as correctly stated by the authors, these discrepancies are due to the different methodologies used and numbers of patients recruited and not to the ascertained diagnosis.  Despite the limited number of study patients, the authors may tentatively suggest guidance on the basis of their results. The latter should be included in the discussion and in the abstract, this highly likely to increase the visibility of the article and of the journal.

The authors have organized this study with the aim of determining the prevalence of lactose intolerance (LI) in 58 patients with Hashimoto’s thyroiditis and assessing the impact of LI on LT4 replacement dose. The cumulative required LT4 dose was not different in patients with or without LI and no significant difference in either TSH values or LT4 dose was observed. The prevalence of LI in patients with HT was 58.6% and did not differ from the global prevalence of LI.

It is a carefully designed and well-presented study. However, I have some concerns:

- Patients with difficulty in controlling their hypothyroidism have more frequent gastrointestinal disorders (GIDs) (McMilan M Drugs RD 2016). The prevalence of LI in a much larger study including 925 patients treated with LT4 was only 7.8%, while gastroesophageal reflux was more frequent, at 33.8%. The study stresses the need for gastrointestinal workup for patients with LT4 malabsorption. Many patients may have other GIDs in addition to LI, such as inflammatory bowel syndrome, which may interfere with LT4 absorption:  this aspect could be more emphatically stated in the study, which shows that nonpersistent LI is not likely to be the cause of LT4 malabsorption, but other underlying GIDs.

- Did the authors detect any difference in LT4 absorption between patients with persistent and nonpersistent lactose intolerance? Munoz-Torres M et al. Thyroid.

- Were all patients treated with LT4 in tablets or else gel formulations?  Some more references can be added: Marek Ruchała et al. Endokrynol Pol 2012; Asik M, Gunes F, et al. Endocrine 2014.

- Of note, intolerance to lactose should be considered in the DD of GIDs that can cause malabsorption of LT (4). The possibility of correcting this disorder with simple dietary measures justifies its consideration.

The results of this study are partially discordant with those of several other studies, as correctly stated by the authors, these discrepancies are due to the different methodologies used and numbers of patients recruited and not to the ascertained diagnosis.  Despite the limited number of study patients, the authors may tentatively suggest guidance on the basis of their results. The latter should be included in the discussion and in the abstract, this highly likely to increase the visibility of the article and of the journal.
